

**Measurement report: Molecular-level investigation of**
**atmospheric cluster ions at the tropical high-altitude research**
**station Chacaltaya (5240 m a.s.l.) in the Bolivian Andes**
Qiaozhi Zha[1], Wei Huang[1*], Diego Aliaga[1], Otso Peräkylä[1], Liine Heikkinen[2], Alkuin
Maximilian Koenig[3], Cheng Wu[4], Joonas Enroth[1], Yvette Gramlich[2], Jing Cai[1], Samara
Carbone[5], Armin Hansel[6], Tuukka Petäjä[1], Markku Kulmala[1,7,8], Douglas Worsnop[1,9],
Victoria Sinclair[1], Radovan Krejci[2], Marcos Andrade[10,11], Claudia Mohr[2], Federico
Bianchi[1*]
*[1] Institute for Atmospheric and Earth System Research /Physics, University of Helsinki,*
*Helsinki, Finland*
*[2] Department of Environmental Science & Bolin Centre for Climate Research, Stockholm*
*University, Stockholm, Sweden*
*[3] Institut des Géosciences de l'Environnement, Univ Grenoble Alpes, CNRS, IRD, Grenoble*
*INP, Grenoble, France*
*[4] Department of Chemistry & Molecular Biology, University of Gothenburg, Sweden*
*[5] Federal University of Uberlândia, Uberlândia, MG, Brazil*
*[6] Institute for Ion and Applied Physics, University of Innsbruck, Innsbruck, Austria*
*[7] Joint International Research Laboratory of Atmospheric and Earth System Sciences, Nanjing*
*University, Nanjing, China*
*[8]Aerosol and Haze Laboratory, Beijing Advanced Innovation Center for Soft Matter Science*
*and Engineering, Beijing University of Chemical Technology, Beijing, China*
*[9] Aerodyne Research, Inc., Billerica, MA, USA*
*[10] Laboratory for Atmospheric Physics, Institute for Physics Research, Universidad Mayor de*
*San Andrés, La Paz, Bolivia*
*[11]epartment of Atmospheric and Oceanic Sciences, University of Maryland, College Park, MD,*
*USA*
*Corresponding author*: Wei Huang, wei.huang@helsinki.fi and Federico Bianchi,
federico.bianchi@helsinki.fi



## Abstract

Air ions are the key components for a series of atmospheric physicochemical
interactions, such as ion-catalyzed reactions, ion-molecule reactions, and ion-induced
new particle formation. They also control atmospheric electrical properties with effects
on global climate. We performed molecular-level measurements of cluster ions at the
high-altitude research station Chacaltaya (CHC; 5240 m a.s.l.), located in the Bolivian
Andes, from January to May 2018 using an atmospheric pressure interface time-of-
flight mass spectrometer. The negative ions mainly consisted of $(H_2SO_4)_{0-3} \cdot HSO_4^-$,
$(HNO_3)_{0-2} \cdot NO_3^-$, $SO_5^-$, $(NH_3)_{1-6} \cdot (H_2SO_4)_{3-7} \cdot HSO_4^-$, malonic acid-derived, and
$CHO/CHON \cdot (HSO_4^-/NO_3^-)$ cluster ions. Their temporal variability exhibited distinct
diurnal and seasonal patterns due to the changes in the corresponding neutral species'
molecular properties (such as electron affinity and proton affinity) and concentrations
resulting from the air masses arriving at CHC from different source regions. The
positive ions were mainly composed of protonated amines and organic cluster ions, but
exhibited no clear diurnal variation. $H_2SO_4$-$NH_3$ cluster ions likely contributed to the
new particle formation process, particularly during wet-to-dry transition period and dry
season when CHC was more impacted by air masses originating from source regions
with elevated $SO_2$ emissions. Our study provides new insights into the chemical
composition of atmospheric cluster ions and their role in new particle formation in the
high-altitude mountain environment of the Bolivian Andes.

## 1  Introduction


Air ions regulate the electrical properties of the atmosphere by serving as carriers of
electrical charges (Williams, 2009). They also play an important role in atmospheric
chemistry by participating/catalyzing ion-molecule reactions and ion-induced new
particle formation (NPF, Hirsikko et al., 2011). The formation of tropospheric ions is
initiated through simple-structured ions, such as $O^+$, $N_2^+$, $O^-$, and $O_2^-$, mainly from
radioactive decay in the soil (e.g., radon and gamma radiation), thunderstorm activity
(lightning), and galactic cosmic rays (GCR). These ions can transfer their charges to
other compounds, leading to the subsequent production of an assortment of ions, such
as the bisulfate ion ($HSO_4^-$), nitrate ion ($NO_3^-$), hydronium ion ($H_3O^+$), and ammonium
ion ($NH_4^+$; Smith and Spanel, 1995; Hirsikko et al., 2011). Depending on their sizes,
air ions are usually classified into cluster ions (diameter ≤ 1.6 nm) that are charged
molecules or molecular clusters, and charged particles (diameter > 1.6 nm; Hirsikko et
al., 2005, 2011; Komppula et al., 2007).
Cluster ions exist almost always in the troposphere and can undergo frequent ion-
molecule reactions during their lifetime (~100 seconds; Manninen et al., 2010; Hirsikko
et al., 2011). Their chemical composition, in addition to the initial ionization, also
depends on the concentrations of the parent neutral species (Eisele, 1986). Bianchi et
al. (2017) showed that the diurnal cycle of negative organic ions followed the variations
of their neutral molecules' concentrations in a boreal forest, since the higher



70 concentrations of neutral molecules would result in a larger probability of them being
71 charged. Moreover, molecular properties of the neutral species, such as electron affinity
72 (EA) and proton affinity (PA), are also important for determining cluster ion
73 composition. Cluster ions derived from molecules with higher EA (e.g., $HSO_4$ and $NO_3$)
74 or PA (e.g., trimethylamine ($C_3H_9N$) and pyridine ($C_5H_5N$)) tend to obtain the ambient
75 negative or positive charge, respectively (Ferguson and Arnold, 1981; Hirsikko et al.,
76 2011). Because of the strong EA or PA, it is almost unlikely that the ions derived from
77 those molecules will further transfer their charges to other neutral compounds via ion-
78 molecule reactions. Thus, these negative ($HSO_4^-$ and $NO_3^-$) and positive ($C_3H_{10}N^+$ and
79 $C_5H_6N^+$) ions are usually more abundant than other ions in the atmosphere (Eisele, 1986;
80 Ehn et al., 2010; Bianchi et al., 2017; Frege et al., 2017). In contrast, charge transfer
81 occurs more easily for ions derived from neutral species of lower EA or PA.

82 Atmospheric cluster ions can contribute to new particle formation (NPF) via ion-
83 induced nucleation (Yu, 2010). Since the discovery of this mechanism in the first cloud
84 chamber study in the early 1900s (Wilson, 1911), ion-induced nucleation has been
85 known as an important source of atmospheric aerosol particles. Recently, a series of
86 chamber studies conducted at the CLOUD (Cosmics Leaving Outdoor Droplets) facility
87 at CERN (the European Centre for Nuclear Research) have shown that aerosol
88 nucleation rates are substantially enhanced in the presence of some specific cluster ions,
89 such as sulfuric acid – ammonia ($H_2SO_4 – NH_3$) cluster ions (Kirkby et al., 2011;
90 Schobesberger et al., 2015), pure $H_2SO_4$ cluster ions (Kirkby et al., 2011), and organic
91 cluster ions (Kirkby et al., 2016). Field measurements have also suggested the important
92 role of atmospheric ions in ion-induced nucleation (Manninen et al., 2010; Hirsikko et
93 al., 2011; Rose et al., 2018; Jokinen et al., 2018; Yan et al., 2018; Beck et al., 2021).
94 Among them, the onsets of high-altitude NPF events, compared to those occurring in
95 the lower troposphere, are often associated with more abundant cluster ions (Lee et al.,
96 2003; Venzac et al., 2008; Boulon et al., 2010). Such increases are due to the higher
97 GCR intensity and lower condensation sink (CS) in the high-altitude regions. As a result,
98 potentially larger contributions of cluster ions to aerosol formation would be expected
99 (Smith and Spanel, 1995; Hirsikko et al., 2011).

100 However, the molecular-level understanding of ambient cluster ions and their influence
101 on NPF in high-altitude environments (in the troposphere) is still very limited. Two
102 mountaintop studies in the Alps show that, depending on the air mass origins, NPF
103 could be triggered by sulfuric acid-ammonia clusters, or nitrate (or sulfuric acid)
104 clustering with highly oxygenated organic molecules (Bianchi et al., 2016; Frege et al.,
105 2017). Another study in the Himalayas found that NPF was mainly driven by organic
106 vapors of biogenic origin (Bianchi et al., 2021). Recently, frequent and intensive NPF
107 events were observed at the high-altitude research station Chacaltaya (CHC; 16.3505°
108 S, 68.1314° W; 5240 m a.s.l.) located in the Bolivian Andes (Rose et al., 2015), but the
109 exact mechanism and the role of cluster ions in aerosol nucleation process remain
110 unclear. Therefore, a detailed investigation of cluster ions at CHC, including their





molecular composition, temporal variation (diurnal and seasonal), and source regions,
is needed in order to understand their role in atmospheric processes such as NPF in the
study regions.
Here we present measurements of atmospheric ions from January to May 2018 at CHC.
The dataset is part of the Southern hemisphere high ALTitude Experiment on particle
Nucleation And growth (SALTENA) field experiment campaign (Bianchi et al., 2022).
During the study period, the sampled air masses originated from various source regions,
such as the Amazon Basin to the east and the Altiplano and the Pacific Ocean to the
west (Fig. 1a; Aliaga et al., 2021). Temporal evolution (diurnal and/or seasonal
variations) of both negative and positive ion composition are investigated, and their
potential connections with source regions and NPF are discussed. Our study thus adds
important observational information on a better understanding of atmospheric ions and
provides new insights into their role in high-altitude NPF in the troposphere of the
Bolivian Andes.

## 125   2   Methods

### 126   2.1 Measurement site description

The high-altitude research station CHC is ~140 m below the summit of Mount
Chacaltaya (5380 m a.s.l.) with an open view to the south and west (Andrade et al.,
2015). The La Paz – El Alto metropolitan area (with 1.7 million inhabitants) is ~1 – 1.6
km lower (in altitude) and ~15 km south of CHC (Fig. 1b). The seasonal meteorological
conditions at CHC depend on the cycle between the wet (November to March; wet-to-
dry transition period in April) and dry (May to September; dry-to-wet transition period
in October) seasons driven by large-scale tropical circulation (Rose et al., 2015; Bianchi
et al., 2022). This pattern also affects the source regions of air masses arriving at CHC
(Aliaga et al., 2021). Additionally, due to the strong diurnal cycle of the planetary
boundary layer (PBL) height and the thermally-induced winds in the mountainous
terrain, CHC is often affected by polluted PBL transported from the La Paz – El Alto
metropolitan area during daytime (Wiedensohler et al., 2018) whereas at night CHC is
located in the residual layer or tropical free troposphere (Coen et al., 2018).





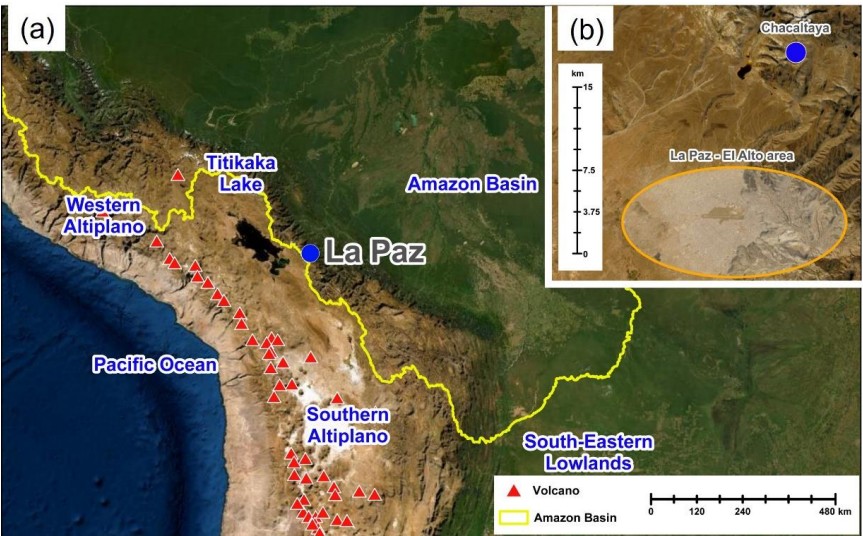


Figure 1 **(a)** True-color satellite image showing the location of CHC (blue circle) and its surrounding

area. The yellow line presents the boundary of the Amazon Basin. Red triangles denote the volcanoes in

this area. **(b)** A zoomed-in true-color satellite image showing the distance between CHC and the La Paz

– El Alto metropolitan area (orange circle). Image sources: Esri, DigitalGlobe, GeoEye, i-cubed, USDA

FSA, USGS, AEX, Getmapping, Aerogrid, IGN, IGP, swisstopo, and the GIS User Community.

## 2.2 Instrumentation

All the instruments involved in this study were installed in a temperature-controlled

measurement room (at ~25 °C). All data are reported in local time (UTC-4).

### 2.2.1   Measurements of atmospheric cluster ions

The composition of cluster ions was measured by an atmospheric pressure interface

time-of-flight mass spectrometer (APi-TOF, Aerodyne Research Inc. & Tofwerk AG).

The APi-TOF consists of an atmospheric pressure interface (APi) module and a time-

of-flight (TOF) mass spectrometer. The APi module allows the instrument to sample

ions from ambient air directly. The positive or negative ions within the sampled airflow

are focused and guided by two quadrupoles and an ion lens, with a gradually decreasing

pressure (from atmospheric pressure to ~$10^{-4}$ mbar), before entering the TOF mass

spectrometer (~$10^{-6}$ mbar). A more detailed description of this instrument is given in

Junninen et al. (2010). In this study, ambient air was sampled through a ~1.5 m stainless

steel tube with a total sample flow of 14 standard liters per minute (SLPM) to ensure

laminar flow during sampling, and 0.8 SLPM of the total flow entered the APi-TOF.

During the wet season, the APi-TOF was first operated in negative mode to measure

negative cluster ions (January), and then switched to positive mode to measure positive

cluster ions (February to March). During the wet-to-dry transition period (April) and



dry season (May), the instrument was changed back to negative mode to investigate the
potential seasonality of negative ion composition. It is important to note that, similar to
Frege et al. (2017), for better characterization of the connection between cluster ions
and NPF, we only included the ion data from cloud-free days in this study (to avoid
influence from, e.g., lightning activity).

### 2.2.2  Measurements of $H_2SO_4$ and oxidized organic molecules

Concentrations of $H_2SO_4$ and oxidized organic molecules (OOM) were measured using
a nitrate ion ($NO_3^-$) based chemical ionization atmospheric pressure interface time-of-
flight mass spectrometer (CI-APi-TOF, Aerodyne Research Inc. & Tofwerk AG;
Jokinen et al., 2012). The instrument is a combination of the APi-TOF and a chemical
ionization (CI) unit, which has been widely used to measure $H_2SO_4$ in the atmosphere
(Jokinen et al., 2012; Bianchi et al., 2016; Zha et al., 2018). In this study, a soft X-ray
source (L9490, Hamamatsu) was used to charge nitric acid ($HNO_3$) in a sheath flow of
20 SLPM to produce the reagent ion, $NO_3^-$. $H_2SO_4$ and OOM in the sample flow (10
SLPM) were then charged by either proton transfer or the formation of an adduct with
the reagent ion during the ~200 ms residence time in the CI unit. A calibration factor of
$1.5 \times 10^{10}$ cm$^{-3}$ for $H_2SO_4$ was determined (with sampling loss corrected) according to
the approach by Kürten et al. (2012). The same calibration coefficient was adopted for
determining OOM concentrations in this study, which could result in an
underestimation of their concentrations due to a lower charging efficiency of OOM than
$H_2SO_4$ by $NO_3^-$ (Hyttinen et al., 2015).

### 2.2.3  Auxiliary measurements

The number concentration and size distribution of atmospheric ions and neutral
particles were measured with a neutral cluster and air ion spectrometer (NAIS, Airel
Ltd., Mirme and Mirme, 2013). The instrument can detect air ions with a diameter from
1.4 to 50 nm, including both cluster ions and charged particles. The details of the
instrument used can be found in Rose et al. (2017).
Particle number size distributions between 10 to 500 nm were measured by a Mobility
Particle Size Spectrometer (MPSS; Wiedensohler et al., 2012), and the data was used
for calculating the CS, which represents the loss rate of condensing vapors and cluster
ions on pre-existing particles (Kulmala et al., 2001).
Meteorological parameters, such as temperature, relative humidity (RH), and global
radiation, were also measured simultaneously at CHC. Detailed descriptions can be
found in Bianchi et al. (2022).

### 2.3 Simulation of air mass origin and history

To understand the source regions and transport pathways of the air masses arriving at
CHC, we used the results of air mass history analysis obtained from FLEXPART-WRF
simulations described in Aliaga et al. (2021). In brief, a Lagrangian transport and



dispersion model (FLEXPART-WRF; version 3.3.2; Brioude et al., 2013) was used to
calculate the air mass history during the campaign period. The backward simulation
was driven by the high spatial and temporal resolution meteorological output from the
Weather Research and Forecasting model (WRF; version 4.0.3; Skamarock et al., 2019).
In the simulation, twenty thousand particles were continuously released every hour
from a 10 m deep layer (0 – 10 m a.g.l.) at CHC and traced back in the atmosphere for
96 hours. The output of the FLEXPART-WRF is the source-receptor relationship (SRR,
in seconds), which is calculated for each geographical grid cell included in the
simulation. The SRR value depends on the particle's residence time and the number of
particles in the output grid cells. Clustering analysis was conducted by applying a series
of pretreatments (e.g., log-polar grid transformation and grid cell pre-processing) and a
k-means clustering algorithm (Lloyd, 1982) to the calculated SSR dataset (see Aliaga
et al. (2021) for more details).

### 2.3.1   Major air mass pathways

Six air mass pathways (PW) representing air masses arriving at CHC were determined
from the clustering analysis. They are named based on their clock positions from CHC
(e.g., 03_PW indicates the pathway with its centroid located at the 3 o'clock direction
(east, 90°) of CHC, Fig. 2a). Characteristics of these air mass pathways, such as source
region, transport distance, and transport time, were distinct from each other (Table 1).
A detailed description of the air mass pathways and their characteristics can be found
in Aliaga et al. (2021).
The influence of each air mass pathway on CHC varied with time, and was estimated
by its SRR percentage ($SRR[\%]_{pathway}$) as in equation (1):

$$SRR[\%]_{pathway} = \frac{SRR_{pathway}}{SRR_{total}} \times 100 \qquad (1)$$

where $SRR_{pathway}$ and $SRR_{total}$ are the residence time of a specific air mass pathway and
in total (96 hours = 345600 seconds) in the simulation, respectively.
Table 1. Overview of the six air mass pathways extracted from Aliaga et al. (2021).

| Pathway | Direction to CHC | Representative source region | Transport distance (km)[1] Median (25 – 75 %) | Transport time (hour) Median (25 – 75 %) |
|---|---|---|---|---|
| **03_PW** | East | Amazon Basin and Eastern/South-Eastern Lowlands | 518 (413–608) | 51 (45-57) |
| **05_PW** | South and Southeast | South-Eastern Lowlands and Southern Altiplano | 428 (303-567) | 45 (36-52) |
| **07_PW** | Southwest | The Pacific Ocean, coastal area, Western Altiplano, and La Paz – El Alto | 721 (577-896) | 54 (45-61) |
| **08_PW** | West | Western Altiplano and | 238 (198-279) | 36 (29-43) |



| | | Titicaca lake, coastal area | | |
|---|---|---|---|---|
| **11_PW** | North and Northwest | Amazon Basin, Western Altiplano, coastal area | 465 (326-563) | 53 (46-59) |
| **12_PW** | North | Amazon Basin | 76 (49 -95) | 27 (21-33) |

[1]Distance between CHC and each pathway's center point (see Fig. 2a).

## 2.3.2 Identification of representative periods for each air mass pathway

Air mass history analysis shows that the air sampled at CHC was typically a mixture of multiple pathways. Thus, the cluster ion composition observed during the study period was often influenced by multiple source regions concurrently. To characterize the influence of every single pathway on cluster ion composition, periods when an air mass pathway exerted its largest impact on CHC (the highest 10% of its SRR[%]$_{pathway}$ values; Fig. 2b) during the whole study period are identified as the representative periods of the specific pathway. For instance, the representative periods of 03_PW (covering, e.g., the Amazon Basin) are more frequently seen during wet season (highest in January), whereas 08_PW (covering, e.g., Altiplano region) has most of its representative periods in dry season (highest in May). Note that SRR[%]$_{pathway}$ of any individual pathway rarely reached 40 % during the whole study period (see Fig. S1), and thus the representative periods cannot be directly identified via SRR[%]$_{pathway}$ values (e.g., using a certain threshold of the value) as in a previous study (Koenig et al., 2021).

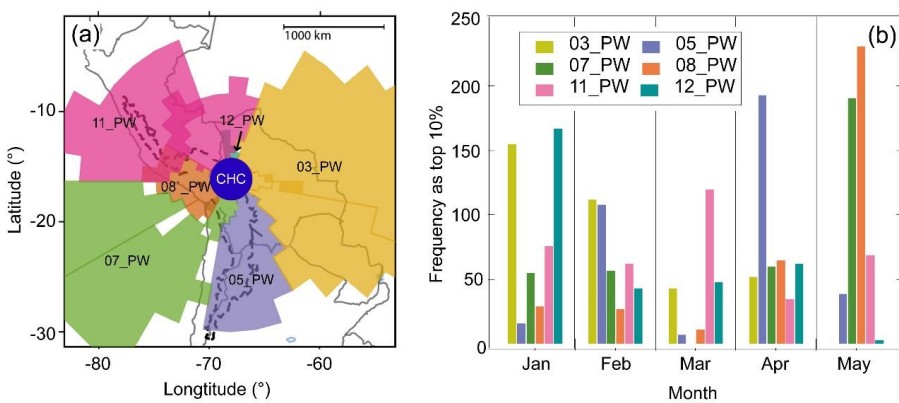

Figure 2 Influence of the six air pathways on CHC from January to May 2018. **(a)** Horizontal profile of the air mass pathways, adapted from Aliaga et al. (2021). **(b)** Frequency of the representative periods for each pathway (the highest 10% of their corresponding SRR[%]$_{pathway}$) in different months.

## 3 Results and discussion

### 3.1 Variation in total ion count

During the study period, the total ion count (TIC) observed by APi-TOF at CHC varied



between <10 and 100 counts per second (cps; Fig. 3 and Fig. S2). Similar variations in
the TIC were also observed in the long-term cluster ion measurements at the high-
altitude station Jungfraujoch in Switzerland (JFJ; 3454 m a.s.l.; Frege et al., 2017) and
were attributed to the seasonal changes of ion precursor and sink. Like the winter–
summer seasonality of the JFJ, CHC and its adjacent mountain areas are frequently
covered by snow in wet season and mostly free of snow in dry season (Bianchi et al.,
2022; Koenig et al., 2021). Thus, in contrast with the generally stable GCR flux
(primarily controlled by the decadal scale solar cycle; Shuman et al., 2015), a reduced
radioactive decay from the soil and a lower ion production rate could be expected at
CHC in wet season than in dry season.
However, the TIC measured by APi-TOF was significantly higher in wet season ($41 \pm$
$23$ cps, mean $\pm$ standard deviation) and the wet-to-dry transition period ($56 \pm 32$ cps)
than in dry season ($14 \pm 11$ cps; Fig. S2). Considering the slight negative correlation
(Pearson's correlation coefficient ($R$): –0.41; Fig. 3) between the TIC and CS
(representing the loss rate of condensing vapors and cluster ions on pre-existing
particles; Kulmala et al., 2001), the observed TIC fluctuation may be related (at least
partially) to the varying CS ($\sim 1 \times 10^{-4}$ s$^{-1}$ in wet season to $\sim 5 \times 10^{-2}$ s$^{-1}$ in the dry season).
Moreover, the cluster ions measured by APi-TOF usually account for only a small
fraction of the total atmospheric ions (Rose et al., 2018). Changes in the fraction of
small ions in total atmospheric ions can potentially lead to a fluctuation in TIC (Frege
et al., 2017). This is illustrated in Figure 3 that a smaller TIC determined from APi-TOF
is associated with a lower fraction of smaller ions (< 2 nm) observed by NAIS (mostly
cluster ions). However, for better characterization of the influences of different ion
composition on CHC and their diurnal and seasonal relative changes, we normalized
the observed ion signal to the TIC for APi-TOF measurements.





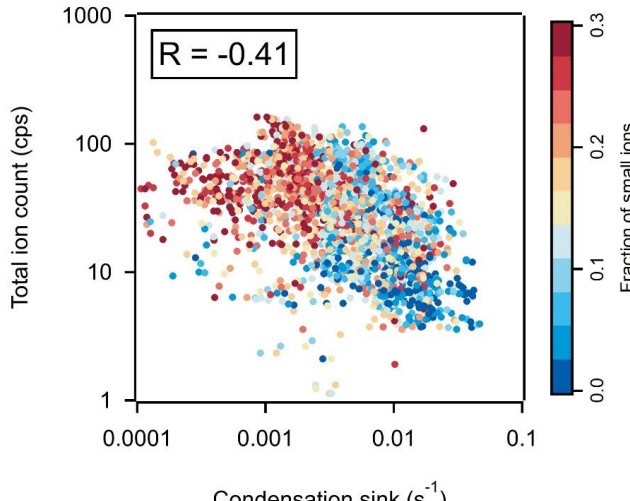


Figure 3 Correlation between the TIC measured by APi-TOF and condensation sink, colored by the
fraction of small ions (defined as concentrations of ions with diameter <2 nm to the total ion
concentrations) determined from NAIS data. Data are shown in the time resolution of 1 hour.
**3.2 Negative ions**
**3.2.1  Main negative ions and their diurnal variation**
A number of negative ions were consistently observed at CHC throughout the study
period (i.e., in the wet, wet-to-dry transition, and dry seasons). Based on the chemical
composition of the observed negative ions, we classified them into eight groups as
follows: sulfuric acid $((H_2SO_4)_{0-3} \cdot HSO_4^-)$, nitric acid $((HNO_3)_{0-2} \cdot NO_3^-)$, $SO_5^-$, sulfuric
acid-ammonia $((NH_3)_{1-6} \cdot (H_2SO_4)_{3-7} \cdot HSO_4^-)$, malonic acid-derived (MA-derived;
including $C_3H_3O_4^-$, $C_3H_4O_4 \cdot NO_3^-$, and $C_3H_4O_4 \cdot HSO_4^-$), oxidized organic molecules
$(CHO/CHON \cdot (HSO_4^-/NO_3^-))$, others (other identified negative ions, such as $IO_3^-$), and
unidentified ions. The campaign-average diurnal variations of these eight negative ion
groups are shown in Figure 4. Different diurnal patterns of each negative ion group
were observed, mainly due to changes in concentrations of the parent neutral species
and their unique physicochemical properties (e.g., the EA and PA of molecules;
Ferguson and Arnold, 1981; Bianchi et al., 2017; Hirsikko et al., 2011).
$(HNO_3)_{0-1} \cdot NO_3^-$ and $(H_2SO_4)_{0-3} \cdot HSO_4^-$ were among the highest in the signal of all
negative ion groups in all seasons, making up 37 % (whole day) and 20 % (daytime;
07:00 – 19:00, and hereafter) of negative ions at CHC during the study period,
respectively. The EA of their neutral molecules ($HSO_4$ and $NO_3$) is higher than that of
most of the neutral species in the atmosphere, and thus hinders the direct electron
transfer from $HSO_4^-$ and $NO_3^-$ to other molecules through ion-molecule reactions
(Ferguson and Arnold, 1981). As a result, these ion groups were found to dominate





negative cluster ions at CHC and also other locations, such as a number of remote sites
in the United States (Eisele, 1986), a boreal forest site in Finland (Ehn et al., 2010;
Bianchi et al., 2017), and the JFJ in Switzerland (Frege et al., 2017).
Distinct diurnal patterns were observed for the $(HNO_3)_{0-1} \cdot NO_3^-$ and $(H_2SO_4)_{0-3} \cdot HSO_4^-$
ion groups. $(HNO_3)_{0-1} \cdot NO_3^-$ exhibited a relatively flat diurnal pattern (see Fig. 4) with
similar fractions at daytime (35 %) and nighttime (19:00 – 07:00; 39 %). Such a diurnal
pattern could result from the high EA of the $NO_3$ molecule (4.01 eV and an additional
~1 eV per $HNO_3$; Ferguson and Arnold, 1981), and its relatively abundant (usually
several ppbv) parent neutral species (e.g., $HNO_3$ and $N_2O_5$) with multiple sources in the
atmosphere (e.g., anthropogenic emission and lightning; Martin et al., 2007). In contrast,
$(H_2SO_4)_{0-3} \cdot HSO_4^-$ exhibited a strong diurnal variation. While the fraction of $(H_2SO_4)_{0-3} \cdot HSO_4^-$ remained low (2 %) during nighttime, it started to increase after sunrise (shortly
after 07:00) and reached a maximum (30 %) at around 10:00. Despite an EA comparable
to that of the $NO_3$ molecule (4.75 eV for $HSO_4$; Wang et al., 2000), the strong diurnal
variation of $(H_2SO_4)_{0-3} \cdot HSO_4^-$ is a result of the photochemical production of neutral
$H_2SO_4$. The influence of neutral $H_2SO_4$ on $(H_2SO_4)_{0-3} \cdot HSO_4^-$ is indicated by their
similar diurnal patterns ($R$: 0.52; see Fig. S3a). Similarly, a higher level of $(NH_3)_{1-6} \cdot (H_2SO_4)_{3-7} \cdot HSO_4^-$ was only observed with the presence of abundant $(H_2SO_4)_{0-3} \cdot HSO_4^-$
during daytime. It is also important to note that the decreases of $(H_2SO_4)_{0-3} \cdot HSO_4^-$ and
$(NH_3)_{1-6} \cdot (H_2SO_4)_{3-7} \cdot HSO_4^-$ at around noontime (12:00; see Fig. 4) coincided with an
enhanced CS, indicating the influence of a higher ion sink in addition to the decrease
in neutral $H_2SO_4$ concentration (Boulon et al., 2010; Frege et al., 2017).





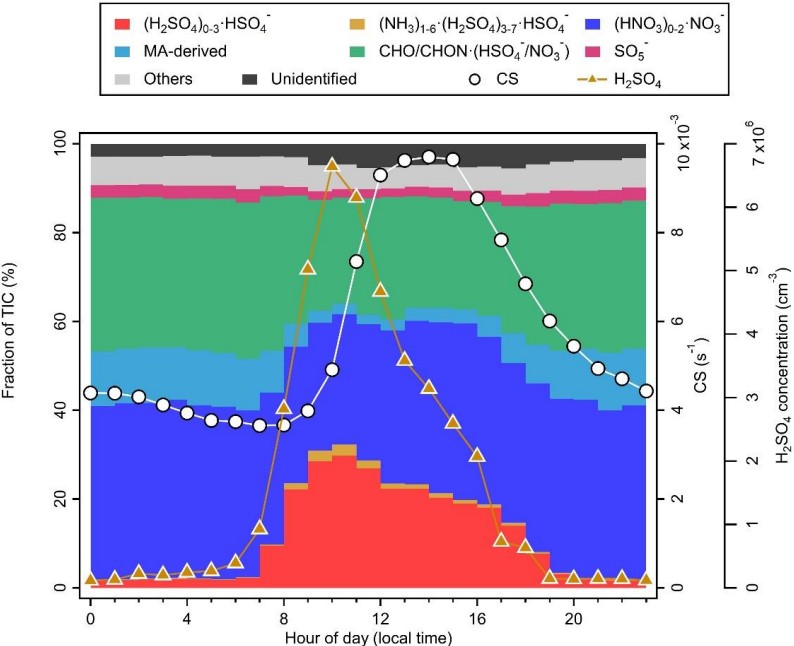


Figure 4 Diurnal variation of the negative ions (fraction of TIC), CS, and neutral $H_2SO_4$ concentrations
at CHC, averaged over the periods when negative ions were measured (i.e., January, April, and May
2018).

The MA-derived ion group is distinguished from the CHO/CHON·($HSO_4^-$/$NO_3^-$) ion
group due to its abundance in total organic ions. This group of ions, mainly composed
of $C_3H_3O_4^-$, is formed from the deprotonation of malonic acid, with a higher EA (~4.60
eV) than that of the $NO_3$ molecule (4.01 eV; Ravi Kumar et al., 2005). Thus, the fraction
of the MA-derived ion group was high during nighttime (12 %; see Fig. 4). In contrast,
its fraction decreased significantly (to 5 %) during daytime due to the increase of
($H_2SO_4$)$_{0-3}$·$HSO_4^-$, which has an even higher EA (4.75 eV for $HSO_4$; Wang et al., 2000).

The CHO/CHON·($HSO_4^-$/$NO_3^-$) ion group, with an overall molecular formula of $C_{2-15}H_{2-26}O_{2-13}N_{0-2}$·$NO_3^-$/$HSO_4^-$, constituted a significant fraction of negative ions (31 %)
at CHC. These organic ions are formed through the adduction between primary charge
carriers, such as $HSO_4^-$ and $NO_3^-$, and neutral OOM. These OOM are likely the
oxidation products of the volatile organic compounds (VOC) from the Amazon and the
adjacent La Paz – El Alto metropolitan area. While the diurnal variation was relatively
small (34 % for the nighttime and 27 % for the daytime), the ion composition of
CHO/CHON·($HSO_4^-$/$NO_3^-$) could be significantly different between daytime and
nighttime due to the availability of the charging ions (see more discussions in Section
3.2.2). A previous study from a boreal forest shows that organic ions are mainly





composed of CHO/CHON·$NO_3^-$ during nighttime, and that the fraction of
CHO/CHON·$HSO_4^-$ increases with the $HSO_4^-$ signal during daytime (Bianchi et al.,
2017). This is also shown by the slightly positive correlation between the
CHO/CHON·$HSO_4^-$ signal fraction and the total neutral OOM concentration during
daytime ($R$: 0.25; see Fig. S3b), whereas no clear dependence was found between
CHO/CHON·($HSO_4^-$/$NO_3^-$) and the total neutral OOM concentration.
The $SO_5^-$ ion group, consisting of $SO_5^-$ ions and/or $O_2$·$SO_3^-$ cluster ions (Bork et al.,
2013; Frege et al., 2017), exhibited a lower fraction (<5 %) than the aforementioned
ion groups during the study period. Similar to that of the $(HNO_3)_{0-1}$·$NO_3^-$ ion group, no
diurnal pattern was evident for the $SO_5^-$ ion group. This may be the result of its different
major formation pathways during daytime and nighttime (Bork et al., 2013; Frege et al.,
2017). Daytime production of $SO_5^-$ ions is likely associated with photo-oxidation of
$SO_2$ (similar to the formation pathway of $H_2SO_4$; Ehn et al., 2010; Schobesberger et al.,
2015). This is shown in the positive correlation ($R$: 0.46 for daytime data in Fig. S3c)
between the neutral $H_2SO_4$ concentration and the signal fraction of the $SO_5^-$ ion group.
During nighttime, however, the $SO_5^-$ ion group is mainly composed of $O_2$·$SO_3^-$ cluster
ions, which are possibly formed via the oxidation of $SO_2$ with $O_3^-$ (producing $SO_3^-$),
and subsequent addition of $O_2$ (Bork et al., 2013).
**3.2.2 Seasonalities of negative ions**
For better seasonality comparison at high-altitude CHC, we calculated the average mass
spectra of the negative ion groups for each season (Fig. 5 for daytime and Fig. S4 for
nighttime). Distinct seasonalities (wet season, wet-to-dry transition period, and dry
season) were found for the majority of the negative ion groups at CHC, including
$(H_2SO_4)_{0-3}$·$HSO_4^-$, $(NH_3)_{1-6}$·$(H_2SO_4)_{3-7}$·$HSO_4^-$, $SO_5^-$, and organic cluster ions, as shown
in the averaged daytime mass spectra (Fig. 5; more detailed reason will be discussed
below). However, the signals of some other negative ion groups, e.g., MA-derived ions
and $(HNO_3)_{0-2}$·$NO_3^-$, were generally stable (with differences ≤ 20 %) across the seasons.
Such unclear seasonalities can be attributed to the high EA (Ferguson and Arnold, 1981;
Ravi Kumar et al., 2005) and/or the stability of the parent neutral species (Martin et al.,
2007; Kerminen et al., 2000; Bikkina et al., 2021). Similar patterns can also be found
in the average nighttime mass spectra among the seasons (Fig. S4).
$(H_2SO_4)_{0-3}$·$HSO_4^-$ group exhibited much higher contribution in dry season (May) than
in wet season (January) and wet-to-dry transition period (April). The daytime fraction
of $(H_2SO_4)_{0-3}$·$HSO_4^-$ increased continuously from 16 % in wet, 20 % in wet-to-dry
transition period, to 30 % in dry season. The maximum number of $H_2SO_4$ molecules
increased concurrently from 2 to 4 in the cluster ions (i.e., from $(H_2SO_4)_2$·$HSO_4^-$ to
$(H_2SO_4)_4$·$HSO_4^-$). Similar trends were also found for other $H_2SO_4$-related ions, such as
the $(NH_3)_{1-6}$·$(H_2SO_4)_{3-7}$·$HSO_4^-$ and $SO_5^-$ during daytime (Fig. 5).



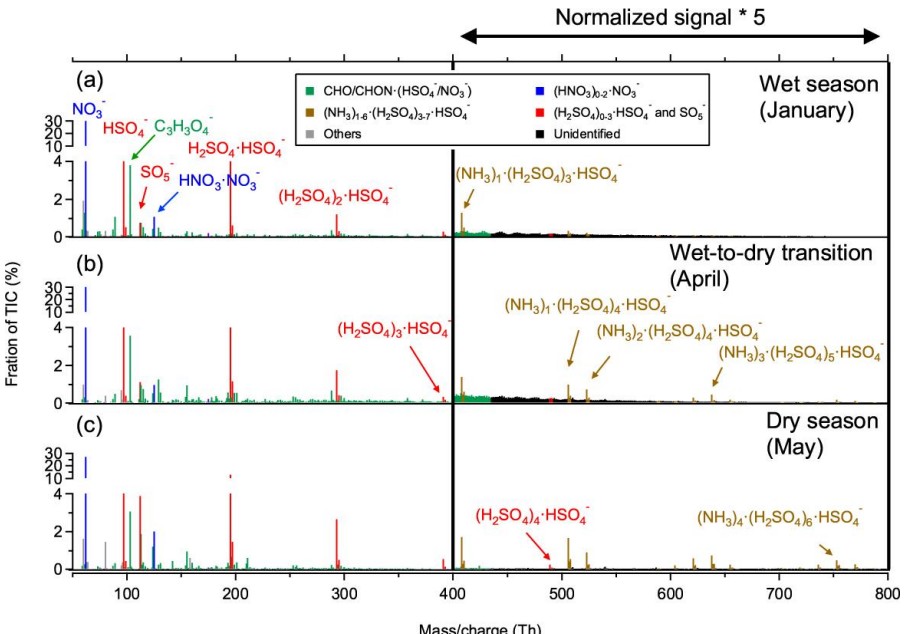

Figure 5 Mass spectra of negative ions at CHC averaged between 07:00 – 19:00 in **(a)** wet season (January), **(b)** wet-to-dry transition period (April), and **(c)** dry season (May). The normalized signal intensities from 400 Th to 800 Th are multiplied by a factor of 5 for better visualization.

The seasonal variations of the aforementioned $H_2SO_4$-related ion groups are likely due to the changes in neutral $H_2SO_4$ (Fig. S5) linked to the changing synoptic-scale wind patterns carrying different air masses with varying $SO_2$ (Bianchi et al., 2022). The air mass pathways 07_PW and 08_PW, covering the Western and Northern Altiplano plateau (see Table 1 and Fig. 2a), where active volcanic degassing of $SO_2$ has been reported (Moussallam et al., 2017; Carn et al., 2017), had their largest influence on CHC in dry season (i.e., May; see Fig. 2). The corresponding daytime fractions of $(H_2SO_4)_{0-3} \cdot HSO_4^-$ from these two pathways (Fig. 6a) were also the highest (27 % and 32 %, respectively). In contrast, air mass pathways 03_PW and 12_PW, originating in the Amazon Basin and Eastern/South-Eastern Lowlands, exerted their most significant impact on CHC in wet season (i.e., January) with lower daytime fractions of $(H_2SO_4)_{0-3} \cdot HSO_4^-$ (13 % and 14 %, respectively). The low fractions of $H_2SO_4$-related cluster ions in wet season are also consistent with the lower $SO_2$ level in the Amazon Basin compared to the Altiplano plateau (Andreae et al., 1990). As for the wet-to-dry transition period (i.e., April), 05_PW covering both the South-Eastern Lowlands and Southern Altiplano plateau (where volcanic degassing is also significant; Carn et al., 2017) had an evident influence on CHC, resulting in a substantial level of $H_2SO_4$-related cluster ions (21 % for daytime). It is also noted that, because of the much lower nocturnal neutral $H_2SO_4$ concentrations, the nighttime fractions of $H_2SO_4$-related cluster ions in all air mass pathways (Fig. 7b) were generally low (< 3 %) and no clear



seasonality was found.
The organic cluster ion group exhibited a distinct seasonal variation than the $(H_2SO_4)_{0-}$
$_3 \cdot HSO_4^-$. The signal fraction of organic cluster ions was higher in wet season (31 % for
daytime and 32 % for nighttime) than in dry season (23 % for daytime and 27 % for
nighttime; Fig. 5 and Fig. S4), but it was highest for the wet-to-dry transition period
(46 % for daytime and 52 % for nighttime; see Fig. 6).
The seasonal changes of organic cluster ions could be due to the combined effect of
different meteorological conditions and VOC from different air mass origins (see Fig.
2). The air masses that originated from the Amazon Basin and Lowlands (03_PW and
12_PW) showed their largest impact on CHC in wet season (i.e., January). They
contained higher fractions of organic cluster ions, which were 35 % and 34 % for
daytime, and 50 % and 45 % for nighttime, respectively (Fig. 6). In dry season (i.e.,
May), however, the changes in air mass origin towards the Altiplano plateau and the
Pacific Ocean led to a lower content of organic cluster ions. The organic cluster ion
fractions for 07_PW and 08_PW (largest influence on CHC in dry season) in May were
23 % and 19 % for daytime, and 29 % and 27 % for nighttime, respectively. As for the
wet-to-dry transition period (i.e., April), due to the combined influences of biogenic
and anthropogenic VOC sources from 05_PW (evident impact on CHC in April),
covering the South-Eastern Lowlands and the Southern Altiplano plateau, the
corresponding organic cluster ion fractions from this air mass pathway were also the
highest (41 % for daytime and 53 % for nighttime; see Fig. 6).

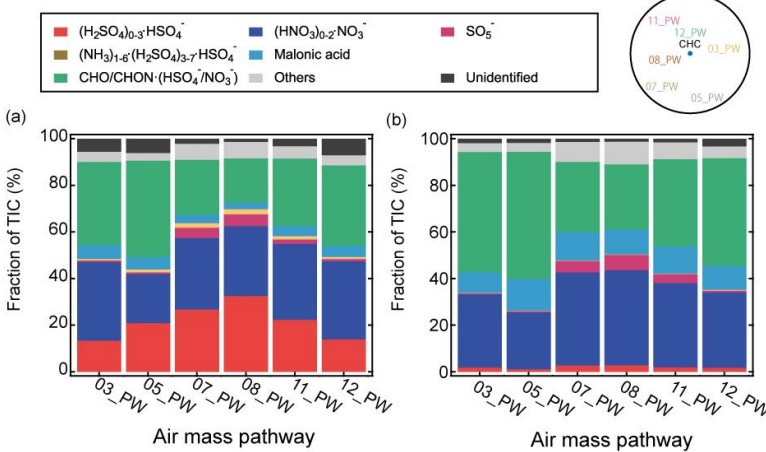


Figure 6 The fractions of the negative ion groups observed at CHC determined during the representative
periods of each air mass pathway (described in Section 2.3.2) for **(a)** daytime (07:00 – 19:00) and **(b)**
nighttime (19:00 – 07:00). A sketch of the horizontal profile of the air mass pathways (Fig. 2a) is shown
in the upper right corner for clarity.



A further investigation of the organic ion group shows that the seasonal trends of the
individual organic ions also varied (Fig. 7 for daytime and Fig. S6 for nighttime).
Whereas the majority of the organic cluster ions at CHC were more abundant during
wet season (Fig. 7a and Fig. S6a), fractions of CHO/CHON·HSO$_4^-$ increased during the
dry season. The observed increases of CHO/CHON·HSO$_4^-$ cluster ions could be
associated with the increased HSO$_4^-$/NO$_3^-$ ratios in dry season (Fig. 5 and Fig. S4).
Similar increases of CHO/CHON·HSO$_4^-$ cluster ions were also found to relate to the
ratio of HSO$_4^-$/NO$_3^-$ in a boreal forest environment (Bianchi et al., 2017). In addition,
changes in OOM composition between wet and dry seasons may also play a role (Fig.
7b and Fig. S6b), as NO$_3^-$ tends to cluster with OOM containing hydroxyl and
hydroperoxyl functional groups (Hyttinen et al., 2015) while some other observed
OOM may be more efficiently charged by HSO$_4^-$.
The seasonal variations of the individual organic cluster ions are likely caused by
different air masses (Fig. 7b and Fig. S6b). The air masses influenced by tropical
rainforest vegetation from the Amazon Basin are dominated by isoprene (C$_5$H$_8$)
emissions and isoprene oxidation products (Bianchi et al., 2022). This region
corresponds to 03_PW and 12_PW (largest impact on CHC in wet season in January)
consisting of relatively higher fractions of organic cluster ions with OOM containing
4-5 carbon atoms (50 % and 46 % for nighttime, and 29 % and 32 % for daytime,
respectively). In contrast, when the air masses were more influenced by the Altiplano
plateau (i.e., 05_PW, 07_PW, and 08_PW, with more anthropogenic emissions and less
vegetation) in wet-to-dry transition period and dry season, organic cluster ions with 6-
8 carbon atoms, potentially originating from anthropogenic sources (e.g., toluene
(C$_7$H$_8$); Huang et al., 2019; Cai et al., 2022), were of higher contributions. The signal
fractions of these organic ions were thus the highest in these air mass pathways,
accounting for 36-39 % for nighttime and 37-39 % for daytime. For all the air mass
pathways, fractions of organic ions with more than 9 carbon atoms were relatively low
(<10 %). This might be due to their lower volatilities compared to OOM with smaller
carbon numbers (Donahue et al., 2012), resulting in a larger probability of them being
removed during their transport to CHC (e.g., condensing on pre-existing particles).



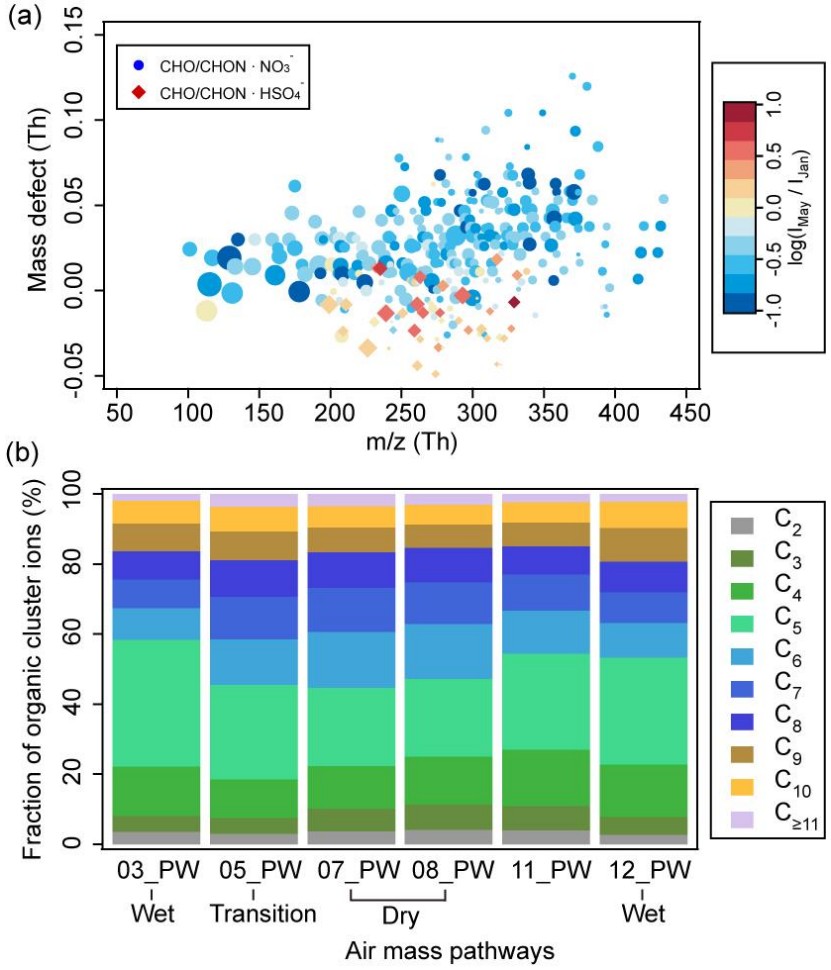

Figure 7 (**a**) Mass defect plot of organic cluster ions during nighttime (19:00-07:00). The color code indicates ratios (in log scale) between median signals of each ion detected in May ($I_{May}$) of dry season and January ($I_{Jan}$) of wet season. The marker size is proportional to the log-transformed median signals of ions in May. (**b**) Fraction of organic cluster ions from different air mass pathways as a function of carbon atom numbers during nighttime (19:00-07:00). A similar figure based on daytime data (07:00-19:00) is in the supplementary information (Figure S5). Note that MA-derived ions were not included in this figure.

## 3.3 Positive ions

Several positive cluster ion groups were consistently observed in February and March (i.e., wet season) during the study period. Based on their chemical composition, the positive cluster ions measured at CHC are classified into four groups (Fig. 8): (1) a series of protonated amines, including trimethylamine ($C_3H_9N \cdot H^+$), pyridine ($C_5H_7N \cdot H^+$), aniline ($C_6H_7N \cdot H^+$), and benzylamine ($C_7H_9 N \cdot H^+$); (2) organic cluster

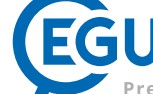



ions consisting of OOM (identified as $C_{3-24}H_{6-39}O_{2-12}N_{0-2}$) clustered with positive
charge carriers such as protons ($H^+$), ammonium ($NH_4^+$), and aminium ($NH^+$) ions; (3)
contamination ions; and (4) unidentified ions (likely organic ions in higher masses;
Bianchi et al., 2021). Contamination in the positive cluster ions includes
ethylhexylglycerin (e.g., $C_{11}H_{24}O_3 \cdot NH^+$), which is widely used in cosmetics (Aerts et
al., 2016), and polydimethylsiloxane (e.g., $(C_2H_6OSi)_7 \cdot NH_4^+$) possibly from instrument
tubing (Bianchi et al., 2014). In contrast to the negative cluster ions, the four positive
cluster ion groups were generally stable with smaller diurnal variability over the study
period (Fig. 9). This is similar to the diurnal patterns determined in previous studies in
a boreal forest environment (Ehn et al., 2010) and at the JFJ (Frege et al., 2017).
However, due to the unavailable measurements of the corresponding neutral species
(e.g., amines), the exact reason for such weak diurnal variations observed in different
locations remains unclear.

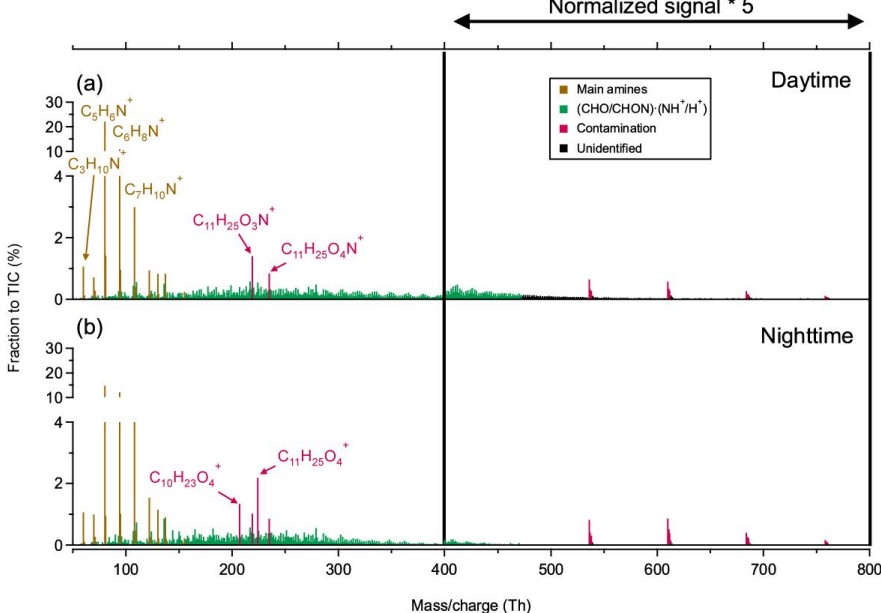

Figure 8 Averaged mass spectra of positive ions at CHC in February and March 2018 when APi-TOF
was operating in positive ion mode (see section 2.2.1), during **(a)** daytime (07:00 – 19:00) and **(b)**
nighttime (19:00 – 07:00). The normalized signal intensities from 400 Th to 800 Th are multiplied by a
factor of 5 for better visualization.
The protonated amines were the most abundant positive ion group (46 %), with no
significant diurnal variations. Nighttime contributions of this ion group (47 %) were
similar to its daytime contributions (45 %; Fig. 9). They also dominated the positive
ion spectra observed in different environments, such as a boreal forest (Ehn et al., 2010),
the JFJ (Frege et al., 2017), and the free troposphere (Schulte and Arnold, 1990). Their



sources have not been fully identified (Kosyakov et al., 2020), but they are widely used
as solvents and dyes (Sims et al., 1989), which may be potential sources of these ions
observed at CHC.

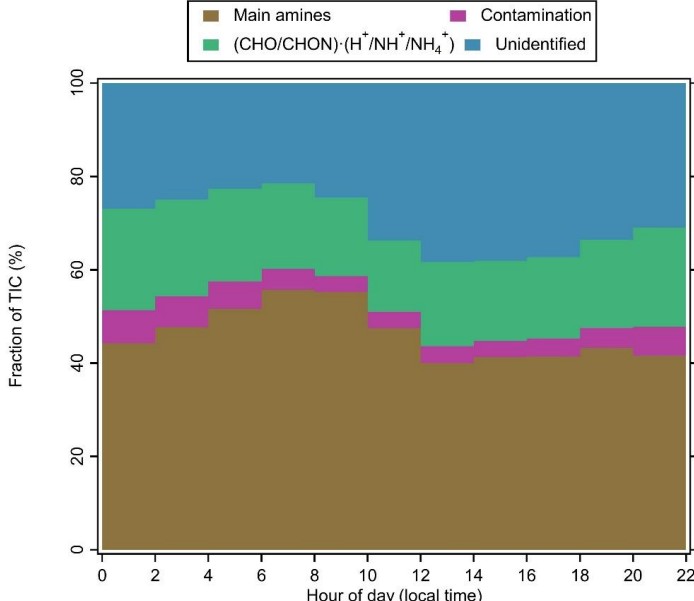

Figure 9 Diurnal variation of positive ion groups at CHC, averaged over measurements in February and
March 2018 (when APi-TOF was operating in positive ion mode, see section 2.2.1).
Positive organic cluster ions were also relatively abundant (19 %) at CHC during the
wet season. Similar to the negative organic ions in wet season (Fig. 6), this reflects the
influence of air masses originating from the Amazon Basin and Eastern/South-Eastern
Lowlands (e.g., 03_PW and 11_PW). Differences in the positive organic ion signals
between nighttime (21 %) and daytime (18 %) were small, which is similar to the
negative organic cluster ions (see Fig. 3). A further investigation of the relationship
between these positive ions and their neutral species is, unfortunately, not possible due
to the unavailability of CI-APi-TOF data in February and March caused by instrumental
issues.
**3.4 Potential connections between atmospheric ions and new particle**
**formation events**
During the SALTENA campaign from January to May 2018, NPF events were
frequently observed at CHC (Fig. S7). While most of them occurred from April (the
wet-to-dry transition period, 21 events) to May (dry season, 26 events), NPF events
seldom occurred during wet season from January to March (8 events in total).



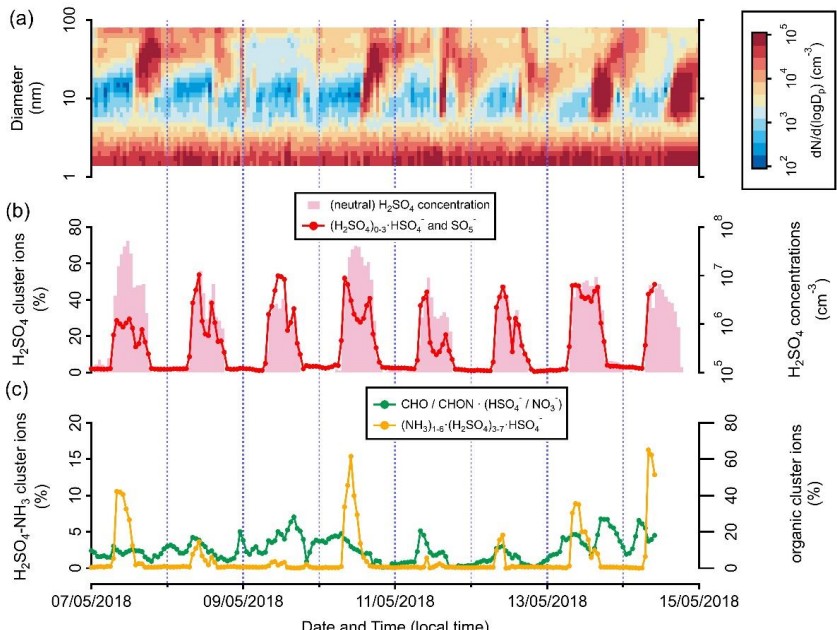


Figure 10 Time series of the **(a)** size distribution of aerosol particles (measured with NAIS and MPSS),
**(b)** signal fraction of the $(H_2SO_4)_{0-3} \cdot HSO_4^-$ ion group and neutral $H_2SO_4$ concentration, and **(c)** signal
fractions of the $(NH_3)_{1-6} \cdot (H_2SO_4)_{3-7} \cdot HSO_4^-$ and negative organic cluster ion groups, observed at CHC
from 7 to 14 May 2018 when NPF occurred frequently.

Previous field studies at high-altitude mountain sites have shown that NPF events can
be triggered by different compounds, such as low-volatile neutral OOM (Bianchi et al.,
2021), neutral $H_2SO_4$ and OOM (Bianchi et al., 2016), and $H_2SO_4$-$NH_3$ cluster ions
(Frege et al., 2017). While the signal fractions of the negative organic cluster ions did
not seem to have a strong correlation with the onset of the NPF events, the fractions of
the $(NH_3)_{1-6} \cdot (H_2SO_4)_{3-7} \cdot HSO_4^-$ (associated with $(H_2SO_4)_{0-3} \cdot HSO_4^-$) always peaked
before NPF events, and started to decrease when NPF started (see example NPF events
on, e.g., 7, 10, 13, and 14 May 2018; Fig. 10).

Moreover, higher levels (up to an order of magnitude) of $(NH_3)_{1-6} \cdot (H_2SO_4)_{3-7} \cdot HSO_4^-$
and $(H_2SO_4)_{0-3} \cdot HSO_4^-$ ions as well as negative organic cluster ions charged by $HSO_4^-$
were also observed during the NPF days (Fig. 11). In contrast, other negative ion groups
(e.g., the majority of the negative organic cluster ions charged by $NO_3^-$) were more
abundant during the non-NPF days. Our observations indicate a potentially important
role of $(NH_3)_{1-6} \cdot (H_2SO_4)_{3-7} \cdot HSO_4^-$ cluster ions in NPF events at CHC from January to
May 2018, particularly in wet-to-dry transition period and dry season.





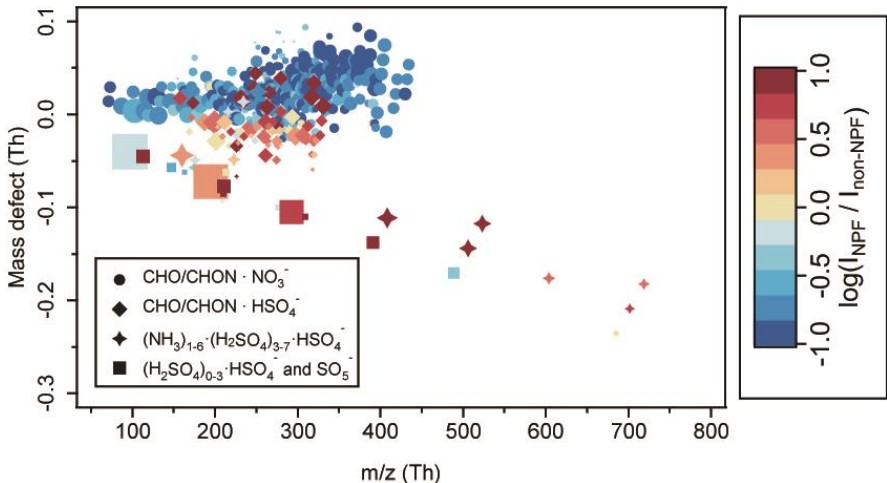


Figure 11 Mass defect plot of differences in negative cluster ion composition between NPF and non-NPF
days. The negative ion composition of NPF events was averaged over all NPF days from 08:00 to 12:00
in January, April, and May 2018 (when APi-TOF was operating in negative ion mode, see section 2.2.1)
at CHC. The ion composition of non-NPF days was averaged over non-NPF days from 08:00 to 12:00
for the same period. The x-axis is the exact mass of cluster ions, and the y-axis is the mass defect. The
color code indicates ratios (in log scale) between median signals of each ion determined in NPF events
($I_{NPF}$) and non-NPF periods ($I_{non-NPF}$). The marker size is proportional to the log-transformed median
signals of ions observed in the NPF events that occurred in January, April and May. Note that $(HNO_3)_{0-2} \cdot NO_3^-$
cluster ions were not included here.
The majority of the observed NPF events occurred when CHC was more impacted by
air masses originating from source regions with elevated $SO_2$ emissions (05_PW,
07_PW, and 08_PW). This is similar to the observations from the high-altitude station
JFJ (Frege et al., 2017). In addition, the fraction of large positive organic cluster ions
(mass range from 500 to 800 Th; Fig. S8) was found to increase during the NPF events
in the wet season. Such large positive organic ions have been found to contribute to
NPF in the Himalayas (Bianchi et al., 2021), and thus, their contribution to NPF can
not be completely ruled out at CHC as well.
## 4. Conclusion
In this study, both negative and positive atmospheric ions were measured at a high-
altitude research station (CHC) in the Bolivian Andes for five months, from January to
May 2018, using an APi-TOF mass spectrometer. Negative ions were mainly composed
of $(H_2SO_4)_{0-3} \cdot HSO_4^-$, $(HNO_3)_{0-2} \cdot NO_3^-$, $SO_5^-$, $(NH_3)_{1-6} \cdot (H_2SO_4)_{3-7} \cdot HSO_4^-$, MA-derived,
and $CHO/CHON \cdot (HSO_4^-/NO_3^-)$ ion groups. Positive ions mainly consisted of a series
of protonated amines $(C_{3-7}H_{7-9}N \cdot H^+)$ and organic cluster ions
$CHO/CHON \cdot (H^+/NH_4^+/NH^+)$. Distinct diurnal variation was observed for the negative
ions, and attributed mainly to the changes in the corresponding neutral species'



concentrations and/or their EA / PA. An example is $H_2SO_4$-related cluster ions, the diel
temporal variation of which was mainly due to the photochemical production of neutral
$H_2SO_4$ during daytime. Strong seasonality of negative ions was also found, such as for
$H_2SO_4$-related cluster ions owing to changes in $SO_2$ and the resulting neutral $H_2SO_4$
concentrations. The seasonal variation was mainly because of the differences in source
regions of air masses arriving at CHC from wet to dry seasons. In contrast, no
significant diurnal variation was observed for the positive ions. The comparison
between NPF and non-NPF days infers that $H_2SO_4$-$NH_3$ cluster ions contribute to the
aerosol nucleation process at CHC, particularly in wet-to-dry transition period and dry
season when CHC was more impacted by air masses originating from source regions
with elevated $SO_2$ emissions. The results further indicate that atmospheric ion
composition at CHC is directly affected by air masses from different source regions.
Measurements of atmospheric ions in the field will improve understanding of
atmospheric physical and chemical processes in the study regions, as the ions play
important roles in atmospheric chemistry through participation in or catalysis of ion-
molecule reactions and ion-induced new particle formation. Our study thus provides
new insights into the chemical composition of atmospheric ions and their potential role
in high-altitude NPF in the Bolivian Andes where both natural (e.g., biogenic and
volcanic) and anthropogenic emissions are important.



**Data availability:** The data that are involved in the figures can be found in doi.org/10.5281/zenodo.7271286 (Zha et al., 2022).

**Author contributions:** Q. Z., W.H., and F.B. analysed the data; D.A. conducted the air mass history analysis; Q.Z., W.H., F.B., D.A., O.P., L.H., A.M.K., C.W., J.E., Y.G., M.A., C.M., and F.B. collected the data and operated the instruments during the measurement campaign. Q.Z. and W.H. wrote the manuscript with contributions from J.C., V.S., S.C., D.W., R.K., M.A., C.M., and F.B. All authors commented on the manuscript.

**Competing interests:** The authors declare no competing interests.

**Acknowledgment:** We thank the Bolivian staff of the IIF-UMSA (Physics Research Institute, UMSA) working at CHC and the long-term observations performed within the framework of GAW and ACTRIS. We thank the IRD (Institut de Recherche pour le Développement) for the logistic and financial support during the campaign, including shipping and customs concerns. We thank the CSC-IT Center for Science, Finland, for the generous computational resources that allowed the WRF and FLEXPART-WRF simulations to be conducted.

**Grant information:** This research has received support from European Union (EU) H2020 program via the findings European Research Council (ERC; project CHAPAs no. 850614 and ATM-GTP no. 742206), the Marie Skłodowska Curie (CLOUD-MOTION no. 764991), the Finnish Centre of Excellence as well as the Academy of Finland (project no. 311932, 315203 and 337549), and the Knut and Alice Wallenberg Foundation (WAF project CLOUDFORM no. 2017.0165).



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
