# Peer review of "Measurement report: Molecular-level investigation of"

_EGUsphere, 2022_

## Referee Comment (RC2)

The paper by Zha et al. presents a study of the chemical composition of naturally charged molecular clusters present at the high altitude site of Chacaltaya, located in the Bolivian Andes. This analysis is mainly based on measurements performed with an APi-TOF during 5 months, in the framework of the SALTENA campaign (Jan-May 2018), and also includes results obtained from the simultaneous deployment of a nitrate CI-APi-TOF at the site. Concerning positive ions, the analysis focuses on the study of the diurnal cycle of the observed signal. For negative ions, on the other hand, the observations conducted over a period of 3 months belonging to distinct seasons (wet, wet to dry transition and dry) allow, in addition to the analysis of the diurnal cycles, the study of seasonal contrasts, in connection with the variability of the origin of the air masses sampled at the site; the involvement of the identified clusters in NPF, which is particularly frequent at the site during the dry season, is also rapidly explored.

This study provides observations of interest regarding the composition of the troposphere and the understanding of specific processes such as NPF; these data are all the more valuable as the process remains globally less well documented at high altitude than at low altitude. I also find the paper very clear and well written. I therefore recommend its publication in ACP. However, I have some questions and suggestions listed below.

P3, L94-95 : I would suggest adding here the review on high-altitude NPF by Sellegri et al. (2019), which includes a section that specifically discusses the role of ions in the process.

P4, L139: The paper by Collaud Coen et al. (2018) investigates the degree of influence of the boundary layer at different high altitude sites but does not specifically address/describe the variability of conditions at the sites in relation to the diurnal cycle of the boundary layer height. These aspects are instead specifically discussed for Chacaltaya in two previous publications: Rose et al. (2015) and Chauvigné et al. (2018).

P6, L167: "we only included the ion data from cloud-free days in this study": I would find it useful to have more information on this filtering process:
- What measurement/method was used? Does the filter only concern clouds detected at the station's altitude, or also clouds that may be above?
- How much data was excluded? Is there a marked variability from month to month?

P6, L188-189: "The instrument can detect air ions with a diameter from 1.4 to 50 nm": two size ranges (corresponding to the two operating modes of the instrument, i.e. naturally charged ions vs total particles) are usually reported (e.g. Manninen et al., 2010: 0.8 – 42 nm for naturally charged ions and ~2 -42 nm for particles), and the range reported here does not correspond to either of the traditionally reported ranges: is the observed difference explained by taking into account the (low) pressure at the site in the calculation of the mobility diameters from the electrical mobilities?

Also, without going into a detailed description of the instrument and its operation, I think it would be useful to say few words about the two measurement modes it allows (naturally charged particles vs. total particles), since the data from these two modes are discussed in the paper and they are in particular characterised by different cut-off diameters due to the presence of charger ions in the "total particles" mode.

P8, L243-244: "the representative periods cannot be directly identified via SRR[%]pathway values (e.g., using a certain threshold of the value) as in a previous study (Koenig et al., 2021).": in order to clarify the explanation, I would suggest indicating briefly why the approach discarded here was possible in the study by Koenig et al. (2021) (longer dataset?).

P12, L328-334 : While both the abundance of the neutral parent species and its EA are discussed to explain the variations observed for $(HNO_3)_{0-1} \cdot NO_3^-$ and $(H_2SO_4)_{0-3} \cdot HSO_4^-$ (and indirectly $(NH_3)_{1-6}(H_2SO_4)_{3-7} \cdot HSO_4^-$) ion groups, only the EA is mentioned in the analysis of the MA-derived ion group. Does this mean that the abundance of malonic acid plays no role in the observed variations?

P12-13, L335-350 and P16, L445-462: Regarding the CHO/CHON·($HSO_4^-$/$NO_3^-$) ion group. The authors highlight both diurnal and seasonal variability (Sect. 3.2.2) in the fraction of negative clusters made up by the CHO/CHON·$NO_3^-$ and CHO/CHON·$HSO_4^-$ groups. In Sect. 3.2.2 the variability in the composition of the OOM (i.e. CHO/CHON) is also discussed but while two figures are presented (Fig. 7.b and S6.b) to distinguish the day/night observations, the analysis is mainly focused on the influence of the different air mass origins, although there are also sometimes marked diurnal contrasts. In particular, a possible link to the day/night contrasted conditions related to the dynamics of the atmospheric boundary layer (as mentioned P4, L135-139) is not discussed, while the study by Beck et al. (2021) shows for example that the negative cluster ions observed in the different atmospheric layers above the boreal forest have different chemical composition. Can the authors comment on this point?

P14, L406: should be Fig. 6b instead of 7b.

P19, Sect. 3.4 :

➔ Yan et al. (2018) showed that there is a priori a link between the number of $H_2SO_4$ molecules present in $H_2SO_4$-$NH_3$ anion clusters and the occurrence of ion induced nucleation in the boreal forest. Did the authors observe, in addition to the link between the occurrence of NPF and the increase in the signal of the $(NH_3)_{1-6}(H_2SO_4)_{3-7}$·$HSO_4^-$ clusters as a whole (Fig. 10.c), such kind of relationship at Chacaltaya? More broadly, in addition to the impact of the identified compounds on the occurrence of NPF, is there an impact on the type (occurrence of growth or only burst, continuous or interrupted growth) and/or intensity (particle formation and/or growth rates) of the events?

➔ As illustrated in Figure S7, and consistent with the previous results of Rose et al. (2015), the highest NPF frequencies are observed during the dry season. Figure 2 shows the frequent arrival of air masses from the Pacific Ocean at the site during this period, again consistent with the results of Rose et al. (2015) who also show a NPF probability close to 100% in these air masses. However, based on this previous work, the observed events seem to be triggered once the air masses arrive on the continent, suggesting that these air masses of marine origin may not directly contain the nucleating species but have a generally favourable character (e.g. low CS) allowing the nucleation of continental species. In Sect. 3.2.1 (P10, L289), the authors quickly mention the identification of $IO_3^-$ but do not mention other compounds of marine origin: are they absent from the spectrum? In view of the results in Fig. S7 and 2, I believe that a more developed discussion of the presence or absence of these marine-derived compounds would be welcome in this study, and would in particular complement the analysis of Rose et al. (2015) regarding the understanding of the favourable character of marine origin air masses for NPF.

P21, L554-556: the comparison with the results obtained on non-event days seems necessary to evaluate the importance of the variability observed on event-days and the associated conclusions.

Fig. 1: I find it confusing that the marker representing Chacaltaya is so close to the "La Paz" indication on panel a. Since panel b is specifically intended to illustrate the positioning of La Paz in relation to the station, I would not include La Paz on panel a to avoid confusion.

Fig. 2: While I understand the general meaning of the results presented in Fig. 2b, I do not understand the unit used on the y axis.

References:

Beck, L. J., Schobesberger, S., Junninen, H., Lampilahti, J., Manninen, A., Dada, L., Leino, K., He, X.-C., Pullinen, I., Quéléver, L. L. J., Franck, A., Poutanen, P., Wimmer, D., Korhonen, F., Sipilä, M., Ehn, M., Worsnop, D. R., Kerminen, V.-M., Petäjä, T., Kulmala, M., and Duplissy, J.: Diurnal evolution of negative atmospheric ions above the boreal forest: from ground level to the free troposphere, Atmos. Chem. Phys., 22, 8547–8577, https://doi.org/10.5194/acp-22-8547-2022, 2022.

Chauvigné, A., Aliaga, D., Sellegri, K., Montoux, N., Krejci, R., Močnik, G., Moreno, I., Müller, T., Pandolfi, M., Velarde, F., Weinhold, K., Ginot, P., Wiedensohler, A., Andrade, M., and Laj, P.: Biomass burning and urban emission impacts in the Andes Cordillera region based on in situ measurements from the Chacaltaya observatory, Bolivia (5240 m a.s.l.), Atmos. Chem. Phys., 19, 14805–14824, https://doi.org/10.5194/acp-19-14805-2019, 2019.

Sellegri, K., Rose, C., Marinoni, A., Lupi, A., Wiedensohler, A., Andrade, M., Bonasoni, P. and Laj, P. : New particle formation : a review of ground-based observations at Mountain research stations, Atmosphere, 10(9), 493, https://doi.org/10.3390/atmos10090493, 2019.

Yan, C., Dada, L., Rose, C., Jokinen, T., Nie, W., Schobesberger, S., Junninen, H., Lehtipalo, K., Sarnela, N., Makkonen, U., Garmash, O., Wang, Y., Zha, Q., Paasonen, P., Bianchi, F., Sipilä, M., Ehn, M., Petäjä, T., Kerminen, V.-M., Worsnop, D. R., and Kulmala, M.: The role of H2SO4-NH3 anion clusters in ion-induced aerosol nucleation mechanisms in the boreal forest, Atmos. Chem. Phys., 18, 13231–13243, https://doi.org/10.5194/acp-18-13231-2018, 2018.

---

## Author Comment (AC1)

We appreciate the detailed comments and insightful suggestions from both reviewers. Our point-by-point responses are listed in green font. The corresponding modifications made to the revised manuscript are highlighted in light blue font. Please note that only references that are part of the replies to the comments are listed in the bibliography at the end of this document. References in copied text excerpts from the manuscript are not included in the bibliography. Line numbers refer to the updated track changes version of the manuscript.

**Reviewer 1 (RC1):**

This manuscript presents a large dataset of atmospheric cluster ions at a high-altitude station in the Sothern Hemisphere over a 5-month campaign, which is important. This work is one of the few existing studies reporting cluster ion composition at high altitudes, and probably the only one in the Southern Hemisphere. The chemical composition of the cluster ions was determined by using state-of-the-art instrumentation. The authors show interesting seasonal variations of the observed ions and attribute them to the properties of parent neutral molecules and different source origins by combing the results from the FLEXPART model. The potential link between the observed cluster ions and aerosol nucleation is also discussed. The manuscript is well-written and within the scope of ACP as a measurement report. I recommend it be accepted after the authors address several (minor) comments listed below:

*Reply*: We thank the reviewer for the positive and constructive suggestions.

1. While the authors attributed the variations of some cluster ions (e.g., NO3- and HSO4-) to the abundance and properties of their parent neutral molecules in a convincing way, it would be better to show the observed concentrations of, e.g., NOx and SO2.

*Reply*: Thanks for the suggestion. We agree with the reviewer that the connection between cluster ions and their parent neutrals can be better demonstrated by showing the concentrations of these neutral species (e.g., $NO_x$ and $SO_2$). Unfortunately, such measurements were unavailable due to instrumental issues during the study. Still, the dependence of cluster ions and their parent neutrals is evident, as we show in Fig. S4 and section 3.2.1 in the manuscript.

2. This reviewer understands that the seasonality of positively charged ions could not be determined because they were only measured in wet season. A significant fraction of the discussion in the manuscript is thus based on measurements of negatively charged cluster ions. However, as mentioned in line 554, the increase of large positive ions was found concurrently with NPF events. It would be better if the authors could specify the chemical composition of the NPF-related positive ions instead of the sum of the signals over a certain mass range.

*Reply*: As suggested by the reviewer, we have added a figure (as Fig. S10) showing the differences in the median contribution of the identified positive organic cluster ions

during two NPF events in February 2018. This provides more detailed information on the chemical composition of the NPF-related positive ions.

Figure S10 is added to the supplementary information and the following sentence is added to the revised manuscript (line 610):

"These organic cluster ions usually contained at least ten carbon atoms (Fig. S10)."

[Figure]

Figure S10 Difference in the identified positive organic cluster ions (median values) observed before (06:00-08:00) and during (10:00-12:00) the NPF events in February 2018 (i.e., 18–19 February). Positive values refer to higher contributions during the NPF events, and negative values higher contributions before the NPF events.

3. It is odd to see the fraction of SA-NH3 and SA cluster ions started increasing before the onset of nucleation (line 530), and an explanation for this may be needed. The reviewer suggests the authors make sure the aerosol data in Fig. 10 are synchronized with cluster ion data.

*Reply*: We thank the reviewer for the detailed comment and suggestion. After a thorough check of the data included in Fig. 10, we find that the size distribution of particles (Fig. 10a) was in UTC time, which is inconsistent with that of the cluster ions and sulfuric acid data (in local time, UTC + 4).

We apologize for this mistake and have updated Fig. 10 and revised the text in the revised manuscript (line 562):

"…always increased concurrently with the number concentration of small particles…"

[Figure]

Figure 10 Time series of the **(a)** size distribution of aerosol particles (measured with NAIS and MPSS), **(b)** signal fraction of the $(H_2SO_4)_{0-3} \cdot HSO_4^-$ ion group and neutral $H_2SO_4$ concentration, and **(c)** signal fractions of the $(NH_3)_{1-6} \cdot (H_2SO_4)_{3-7} \cdot HSO_4^-$ and negative organic cluster ion groups, observed at CHC from 7 to 14 May 2018 when NPF occurred frequently.

---

## Author Comment (AC2)

We appreciate the detailed comments and insightful suggestions from both reviewers. Our point-by-point responses are listed in green font. The corresponding modifications made to the revised manuscript are highlighted in light blue font. Please note that only references that are part of the replies to the comments are listed in the bibliography at the end of this document. References in copied text excerpts from the manuscript are not included in the bibliography. Line numbers refer to the updated track changes version of the manuscript.

**Reviewer 2 (RC2):**

The paper by Zha et al. presents a study of the chemical composition of naturally charged molecular clusters present at the high altitude site of Chacaltaya, located in the Bolivian Andes. This analysis is mainly based on measurements performed with an APi-TOF during 5 months, in the framework of the SALTENA campaign (Jan-May 2018), and also includes results obtained from the simultaneous deployment of a nitrate CI-APi-TOF at the site. Concerning positive ions, the analysis focuses on the study of the diurnal cycle of the observed signal. For negative ions, on the other hand, the observations conducted over a period of 3 months belonging to distinct seasons (wet, wet to dry transition and dry) allow, in addition to the analysis of the diurnal cycles, the study of seasonal contrasts, in connection with the variability of the origin of the air masses sampled at the site; the involvement of the identified clusters in NPF, which is particularly frequent at the site during the dry season, is also rapidly explored.

This study provides observations of interest regarding the composition of the troposphere and the understanding of specific processes such as NPF; these data are all the more valuable as the process remains globally less well documented at high altitude than at low altitude. I also find the paper very clear and well written. I therefore recommend its publication in ACP. However, I have some questions and suggestions listed below.

*Reply*: We thank the reviewer for the very insightful comments and suggestions.

P3, L94-95: I would suggest adding here the review on high-altitude NPF by Sellegri et al. (2019), which includes a section that specifically discusses the role of ions in the process.

*Reply*: Thanks for the suggestion. We have added this reference to the revised manuscript (line 95).

P4, L139: The paper by Collaud Coen et al. (2018) investigates the degree of influence of the boundary layer at different high altitude sites but does not specifically address/describe the variability of conditions at the sites in relation to the diurnal cycle of the boundary layer height. These aspects are instead specifically discussed for Chacaltaya in two previous publications: Rose et al. (2015) and Chauvigné et al. (2018).

*Reply*: Thanks for the suggestion. We have added the corresponding references to the revised manuscript (line 138).

P6, L167: "we only included the ion data from cloud-free days in this study": I would find it useful to have more information on this filtering process:

- What measurement/method was used? Does the filter only concern clouds detected at the station's altitude, or also clouds that may be above?

- How much data was excluded? Is there a marked variability from month to month?

*Reply*: We thank the reviewer for pointing this out, and we apologize for the unclear statement in the manuscript. Based on the method suggested by Rose et al. (2015), the measurement site (or the same altitude) was assumed to be in a cloud when the relative humidity exceeds 95 %. Clouds above the site were not considered in the filtering process. From January to May, the proportions of cloud-free hours to the total measurement time were 72 %, 78 %, 79 %, 98 %, and 98 %, respectively (added as Fig. S1). Such a variation overall agrees with Rose et al. (2015), in which more cloudy days were observed during the wet season at Chacaltaya.

Figure S1 is added to the supplementary information, and we have made the following modification to the revised manuscript (line 168):

"…observed under the cloud-free condition in this study (to avoid influence from, e.g., lightning activity). CHC was considered to be affected by clouds when relative humidity (RH) exceeded 95 %, as suggested by a previous study at the same location (Rose et al., 2015a). From January to May 2018, the proportions of cloud-free hours to the total measurement time were 72 %, 78 %, 79 %, 98 %, and 98 %, respectively (Fig. S1)."

[Figure]

Figure S1 Proportions of cloud-free hours to the total measurement time from January to May 2018.

P6, L188-189: "The instrument can detect air ions with a diameter from 1.4 to 50 nm": two size ranges (corresponding to the two operating modes of the instrument, i.e. naturally charged ions vs total particles) are usually reported (e.g. Manninen et al., 2010: 0.8 – 42 nm for naturally charged ions and ~2 -42 nm for particles), and the range reported here does not correspond to either of the traditionally reported ranges: is the observed difference explained by taking into account the (low) pressure at the site in the calculation of the mobility diameters from the electrical mobilities?

Also, without going into a detailed description of the instrument and its operation, I think it would be useful to say few words about the two measurement modes it allows (naturally charged particles vs. total particles), since the data from these two modes are discussed in the paper and they are in particular characterised by different cut-off diameters due to the presence of charger ions in the "total particles" mode.

*Reply*: Thanks for the detailed suggestion. The size range of the NAIS used in our study was corrected based on a side-by-side comparison with an updated version of NAIS (designed for measurements in high-altitude/low-pressure environments; Mirme et al., (2010)) performed at CHC. As noted by the reviewer, the size range of air ions and particles measured by NAIS has shifted from the typically reported values (0.8 – 42 nm and 2 – 42 nm, respectively) to larger values (1.4 – 50 nm and 3 – 50 nm, respectively).

We have modified the following sentence to explain it more explicitly in the manuscript (line 194):

"…mobility diameters from 1.4 – 50 nm and 3 – 50 nm, respectively. It is important to note that the size range of the NAIS was corrected based on a side-by-side comparison with an updated version of NAIS (designed for measurements under low pressure environments; Mirme et al., (2010)) at CHC. Thus, the size range of detection was slightly different from the traditionally reported ranges (i.e., 0.8 – 42 nm and 2 – 42 nm, respectively; Manninen et al., 2010). "

P8, L243-244: "the representative periods cannot be directly identified via SRR[%]pathway values (e.g., using a certain threshold of the value) as in a previous study (Koenig et al., 2021).": in order to clarify the explanation, I would suggest indicating briefly why the approach discarded here was possible in the study by Koenig et al. (2021) (longer dataset?).

*Reply*: We thank the reviewer for the suggestion. We have modified the following statement in the revised manuscript (line 252):

"…due to the relatively short study period (see Fig. S2), and thus the representative periods cannot be directly identified via SRR[%]pathway values. In contrast, such representative periods were determined by using a certain threshold (e.g., > 70 %) in a previous study at CHC, which was based on a more than 6-year dataset (Koenig et al.

(2021)."

P12, L328-334 : While both the abundance of the neutral parent species and its EA are discussed to explain the variations observed for $(HNO_3)_{0-1} \cdot NO_3^-$ and $(H_2SO_4)_{0-3} \cdot HSO_4^-$ (and indirectly $(NH_3)_{1-6}(H_2SO_4)_{3-7} \cdot HSO_4^-$) ion groups, only the EA is mentioned in the analysis of the MA-derived ion group. Does this mean that the abundance of malonic acid plays no role in the observed variations?

*Reply*: We understand the concern from the reviewer. We have added the following sentences discussing the potential role of neutral malonic acid in determining the MA-derived ion group in the atmosphere to the revised manuscript (line 345):

"Malonic acid, similar to $HNO_3$, has multiple origins in the atmosphere, such as primary and secondary anthropogenic sources, biogenic sources, and the degradation of larger organic compounds (Braban et al., 2003). It is also one of the main dicarboxylic acids which make up a substantial fraction of total carbon in aerosol particles (Kawamura and Bikkina, 2016). However, the concentration of gas-phase malonic acid is less well documented and is estimated to be in the range of $10^7$ to $10^9$ $cm^{-3}$, up to three orders of magnitude higher than the typically reported ambient $H_2SO_4$ concentration (Fang et al., 2020)."

P12-13, L335-350 and P16, L445-462: Regarding the $CHO/CHON \cdot (HSO_4^-/NO_3^-)$ ion group. The authors highlight both diurnal and seasonal variability (Sect. 3.2.2) in the fraction of negative clusters made up by the $CHO/CHON \cdot NO_3^-$ and $CHO/CHON \cdot HSO_4^-$ groups. In Sect. 3.2.2 the variability in the composition of the OOM (i.e. CHO/CHON) is also discussed but while two figures are presented (Fig. 7.b and S6.b) to distinguish the day/night observations, the analysis is mainly focused on the influence of the different air mass origins, although there are also sometimes marked diurnal contrasts. In particular, a possible link to the day/night contrasted conditions related to the dynamics of the atmospheric boundary layer (as mentioned P4, L135-139) is not discussed, while the study by Beck et al. (2021) shows for example that the negative cluster ions observed in the different atmospheric layers above the boreal forest have different chemical composition. Can the authors comment on this point?

*Reply*: Indeed, the chemical composition of OOM is affected by the precursor of OOM (i.e., volatile organic compound; VOC) and oxidant (e.g., ozone, hydroxyl radical, and nitrate radical). This is more complex at Chacaltaya because the major pathway of sampled air varies between wet, wet-to-dry transition, and dry seasons (as shown in Fig. 2). The significantly diverged VOC sources (e.g., biomes shown in Aliaga et al. (2021)) within the changing air pathways are expected to be the primary reason for the variation of the OOM composition. However, as the reviewer also pointed out, the evolution of the atmospheric boundary layer may also lead to changes in daytime/nighttime conditions that subsequently affect the OOM composition (Beck et al., 2022). Thus, we have made the following modifications to the revised manuscript (line 361):

"…the Amazon, the Altiplano, and the adjacent La Paz – El Alto metropolitan area. Their chemical composition is potentially affected by the changing air pathways

covering different VOC source regions (Aliaga et al., 2021), and by the different conditions during daytime/nighttime due to the evolution of the different atmospheric layers (Beck et al., 2022). While the diurnal variation was relatively small (34 % for the nighttime and 27 % for the daytime), the ion composition of CHO/CHON·(HSO$_4^-$/NO$_3^-$) could also…"

P14, L406: should be Fig. 6b instead of 7b.

*Reply*: Corrected.

P19, Sect. 3.4 :

➔ Yan et al. (2018) showed that there is a priori a link between the number of H$_2$SO$_4$ molecules present in H$_2$SO$_4$-NH$_3$ anion clusters and the occurrence of ion induced nucleation in the boreal forest. Did the authors observe, in addition to the link between the occurrence of NPF and the increase in the signal of the (NH$_3$)$_{1-6}$(H$_2$SO$_4$)$_{3-7}$·HSO$_4^-$ clusters as a whole (Fig. 10.c), such kind of relationship at Chacaltaya? More broadly, in addition to the impact of the identified compounds on the occurrence of NPF, is there an impact on the type (occurrence of growth or only burst, continuous or interrupted growth) and/or intensity (particle formation and/or growth rates) of the events?

*Reply*: Thanks for the suggestion. We have added a new figure as Figure 11 showing the connection between the maximum number of H$_2$SO$_4$ observed in the (H$_2$SO$_4$)$_2$·HSO$_4^-$ and (NH$_3$)$_{1-6}$·(H$_2$SO$_4$)$_{3-7}$·HSO$_4^-$ cluster ions during NPF events and the number of NPF days or different NPF classes. This provides more detailed information on the impact of the chemical composition of different ions on the frequency and type of NPF events observed at Chacaltaya.

We have added Figure 11 and the following discussion to the revised manuscript (line 564):

"The number of NPF days increased when more H$_2$SO$_4$ molecules were present in the (NH$_3$)$_{1-6}$·(H$_2$SO$_4$)$_{3-7}$·HSO$_4^-$ cluster ion (Fig. 11a). In particular, more than half of the NPF events (28 out of the total 55 events) were observed in the presence of (NH$_3$)$_{4-6}$·(H$_2$SO$_4$)$_7$·HSO$_4^-$. The majority (35 events) of all the NPF events exhibited clear nucleation and growth processes (i.e., Class 1 events; the classification is defined following the approach by Yli-Juuti et al. (2009); Fig. 11b). In contrast, only Class 2 (similar to Class 1 but with less clarity) and bump events (early growth of the newly formed particles is interrupted) were observed when only (H$_2$SO$_4$)$_2$·HSO$_4^-$ was observed."

[Figure]

Figure 11 Connection between the maximum number of H₂SO₄ (in addition to HSO₄⁻) observed in the (H₂SO₄)₂·HSO₄⁻ and (NH₃)₁₋₆·(H₂SO₄)₃₋₇·HSO₄⁻ cluster ions during NPF events and **(a)** the number of NPF days; **(b)** proportions of different NPF classes.

➔ As illustrated in Figure S7, and consistent with the previous results of Rose et al. (2015), the highest NPF frequencies are observed during the dry season. Figure 2 shows the frequent arrival of air masses from the Pacific Ocean at the site during this period, again consistent with the results of Rose et al. (2015) who also show a NPF probability close to 100% in these air masses. However, based on this previous work, the observed events seem to be triggered once the air masses arrive on the continent, suggesting that these air masses of marine origin may not directly contain the nucleating species but have a generally favourable character (e.g. low CS) allowing the nucleation of continental species. In Sect. 3.2.1 (P10, L289), the authors quickly mention the identification of IO₃⁻ but do not mention other compounds of marine origin: are they absent from the spectrum? In view of the results in Fig. S7 and 2, I believe that a more developed discussion of the presence or absence of these marine-derived compounds would be welcome in this study, and would in particular complement the analysis of Rose et al. (2015) regarding the understanding of the favourable character of marine origin air masses for NPF.

*Reply*: We thank the reviewer for the comment. We have observed the enhanced concentration of methanesulfonic acid (MSA; CH₃SO₃⁻; typically from marine sources, Hodshire et al. (2018)) with the nitrate CI-APi-TOF when air masses originated from the Pacific Ocean, which has been reported by Scholz et al. (2023). Unfortunately, as shown in Fig. R1, we do not find an evident connection between NPF events and the presence/absence of CH₃SO₃⁻ and IO₃⁻ measured by the APi-TOF (compared to (NH₃)₁₋

$_6 \cdot (H_2SO_4)_{3-7} \cdot HSO_4^-$ cluster ions). The time series of $CH_3SO_3^-$ and $IO_3^-$ were also not consistent during most of the measurements, prohibiting us from drawing further conclusions about the marine air-induced NPF. Moreover, generally higher values of the condensation sink were observed in the dry season (May; Fig. R2). This is consistent with the high-altitude study at Jungfraujoch (Boulon et al., 2011) and a previous study at CHC (Rose et al., 2015), in which the condensation sink was found to be positively correlated with NPF frequency. The authors in these previous studies suggested that this is due to the efficient production of condensable gases (e.g., $H_2SO_4$) via photochemical processes during transport, while preexisting particles were diluted.

We have added the following sentences to the revised manuscript:

Line 548: "This is consistent with a previous study performed at CHC (Rose et al., 2015a), which also found that NPF events mainly occurred during dry season."

Line 599: "Moreover, consistent with the previous findings from Rose et al. (2015a), CHC was affected by the frequent arrival of air masses from the Pacific Ocean during the dry season (see also Fig. 2). And the NPF during the dry season at CHC seems to be triggered once the air masses arrive on the continent (Rose et al., 2015). However, these air masses of marine origin may not directly contain the nucleating species (also reflected in the poor connection between NPF events and the levels of methanesulfonic acid ion ($CH_3SO_3^-$) or $IO_3^-$; data not shown). In addition, inconsistent time series of $CH_3SO_3^-$ and $IO_3^-$ prohibit us from drawing further conclusions on the role of marine-derived compounds on NPF at Chacaltaya during the dry season."

[Figure]

Figure R1 Time series of the **(a)** size distribution of aerosol particles (measured with NAIS and MPSS), **(b)** signal fractions of $IO_3^-$ and $CH_3SO_3^-$ cluster ions, **(c)** signal fraction of the $(H_2SO_4)_{0-3} \cdot HSO_4^-$ ion group and neutral $H_2SO_4$ concentration, and **(d)** signal fractions of the $(NH_3)_{1-6} \cdot (H_2SO_4)_{3-7} \cdot HSO_4^-$ and negative organic cluster ion groups, observed at CHC from 7 to 14 May 2018 when NPF occurred frequently.

[Figure]

Figure R2 Monthly condensation sink values at CHC during the study period.

P21, L554-556: the comparison with the results obtained on non-event days seems necessary to evaluate the importance of the variability observed on event-days and the associated

conclusions.

*Reply*: We have added another panel showing the results from non-NPF days to Fig. S9 to support the associated conclusions.

[Figure]

Figure S9 Variations of positive organic cluster ions in different mass ranges averaged during **(a)** NPF days and **(b)** non-NPF days in February and March 2018.

Fig. 1: I find it confusing that the marker representing Chacaltaya is so close to the "La Paz" indication on panel a. Since panel b is specifically intended to illustrate the positioning of La Paz in relation to the station, I would not include La Paz on panel a to avoid confusion.

*Reply*: Modified.

[Figure]

Figure 1 **(a)** True-color satellite image showing the location of CHC (blue circle) and its surrounding area. The yellow line presents the boundary of the Amazon Basin. Red triangles denote the volcanoes in this area. **(b)** A zoomed-in true-color satellite image showing the distance between CHC and the La Paz – El Alto metropolitan area (orange circle). Image sources: Esri, DigitalGlobe, GeoEye, i-cubed, USDA FSA, USGS, AEX, Getmapping, Aerogrid, IGN, IGP, swisstopo, and the GIS User Community.

Fig. 2: While I understand the general meaning of the results presented in Fig. 2b, I do not understand the unit used on the y axis.

*Reply*: The unit for the y axis in Fig. 2b is added.

[Figure]

Figure 2 Influence of the six air pathways on CHC from January to May 2018. **(a)** Horizontal profile of the air mass pathways, adapted from Aliaga et al. (2021). **(b)** Frequency of the representative periods for each pathway (the highest 10% of their

corresponding SRR[%]$_{pathway}$) in different months.

*Reference*:

Rose, C. et al. Frequent nucleation events at the high altitude station of Chacaltaya (5240 m a.s.l.), Bolivia, Atmos. Environ., 102, 18–29 (2015).

Manninen, H. E. et al. EUCAARI ion spectrometer measurements at 12 European sites – analysis of new particle formation events, Atmos. Chem. Phys., 10, 7907–7927 (2010).

Mirme, S. et al. Atmospheric sub-3 nm particles at high altitudes, Atmos. Chem. Phys., 10, 437–451, (2010).

Braban, C. F. et al. Phase Transitions of Malonic and Oxalic Acid Aerosols, J. Phys. Chem. A, 107, 6594–6602 (2003).

Fang, X. et al. Observational Evidence for the Involvement of Dicarboxylic Acids in Particle Nucleation, Environ. Sci. Technol. Lett., 7, 388–394 (2020).

Kawamura, K. and Bikkina, S. A review of dicarboxylic acids and related compounds in atmospheric aerosols: Molecular distributions, sources and transformation, Atmos. Res., 170, 140–160 (2016).

Beck, L. et al. Diurnal evolution of negative atmospheric ions above the boreal forest: from ground level to the free troposphere, Atmos. Chem. Phys., 22, 8547–8577 (2022).

Aliaga, D. et al. Identifying source regions of air masses sampled at the tropical high-altitude site of Chacaltaya using WRF-FLEXPART and cluster analysis, Atmos. Chem. Phys., 21, 16453–16477 (2021).

Yli-Juuti, T. et al. Characteristics of new particle formation events and cluster ions at K-puszta, Hungary. Boreal Env. Res. 14, 683–698 (2009).

Hodshire, A. et al. The potential role of methanesulfonic acid (MSA) in aerosol formation and growth and the associated radiative forcings, Atmos. Chem. Phys., 19, 3137–3160 (2019).

Scholz, W. et al. Measurement report: Long-range transport and the fate of dimethyl sulfide oxidation products in the free troposphere derived from observations at the high-altitude research station Chacaltaya (5240 m a.s.l.) in the Bolivian Andes, Atmos. Chem. Phys., 23, 895–920 (2023).

Boulon, J. et al. New particle formation and ultrafine charged aerosol climatology at a high altitude site in the Alps (Jungfraujoch, 3580 m a.s.l., Switzerland), Atmos. Chem. Phys., 10, 9333–9349 (2010).